# Focused low-intensity hippocampal transcranial ultrasound stimulation (TUS) for sleep disturbances in patients with chronic tinnitus: A study protocol for a pilot randomized controlled trial

Xi Ni[1], Yuk Shan Yuen[1,2], Zeyan Li[1], Kaipeng Wang[1], Natalie Shu Yang[3], Yi Yuan[4,5], Lin Meng[6], Liwei Guo[7], Hanna Lu[1]*

1 Department of Psychiatry, The Chinese University of Hong Kong, Hong Kong SAR, China,
2 Department of Linguistics and Modern Languages, The Chinese University of Hong Kong, Hong Kong SAR, China, 3 Department of Psychology, University of Greifswald, Greifswald, Germany, 4 School of Electrical Engineering, Yanshan University, Qinhuangdao, China, 5 Key Laboratory of Intelligent Rehabilitation and Neuromodulation of Hebei Province, Yanshan University, Qinhuangdao, China, 6 Academy of Medical Engineering and Translational Medicine, Tianjin University, Tianjin, China, 7 Department of Mechanical Engineering, University College London, London, United Kingdom

* hannalu@cuhk.edu.hk

## Abstract

### Background

Sleep disturbances are very common in tinnitus sufferers with a high prevalence ranging from 50% to 77%. Untreated sleep disturbances and tinnitus can cause brain shrinkage and lead to cognitive impairments in late adulthood. Until now, non-pharmacological treatments are very few for older patients suffering from sleep disturbances and chronic tinnitus. Even though clinical trials of transcranial magnetic stimulation (TMS) have shown positive results in the treatment of either sleep disturbances or chronic tinnitus, the results are highly varied due to the superficial cortical target. Compared to TMS, focused low-intensity transcranial ultrasound stimulation (TUS) is a newly developed modality of non-invasive brain stimulation that offers promising therapeutic effects by transmitting acoustic energy into deep brain structures with a high spatial resolution (i.e., sub-millimeter), which sparks interest in managing the comorbidities in ageing populations.

### Methods and design

Chinese individuals between the ages of 60 and 90 years, who are right-handed and have sleep disturbances and chronic tinnitus, will participate in this pilot randomized clinical trial (RCT). Eligible participants will be randomly assigned to two treatment groups (1:1 ratio): low-intensity TUS or sham TUS (i.e., placebo-controlled group).

**Data availability statement:** No datasets were generated or analysed during the current study. All relevant data from this study will be made available upon study completion.

**Funding:** The author(s) received no specific funding for this work.

**Competing interests:** The authors have declared that no competing interests exist.

Each group will consist of 15 participants. Before the treatment, high-resolution T1-weighted magnetic resonance imaging (MRI) data will be used to create a computational head model for each participant. The head model will help identify the treatment target of the left hippocampus. The treatments schedule contains six sessions of low-intensity TUS, three times per week, lasting two weeks. Each session of treatment lasts for 80 seconds. Throughout the study, outcome measurements will be conducted at four time points, including baseline, 2nd week, 6th week, and 12th week. The primary outcomes include subjective sleep quality and severity of tinnitus. The secondary measurements include actigraphy, tinnitus handicap inventory and glymphatic function. Participants' adherence to the program and any adverse event will be closely monitored throughout the duration of the clinical trial.

## Conclusions

It is expected that a 2-week treatment of low-intensity TUS will show significant enhancement in sleep quality and the severity of tinnitus symptoms compared to sham TUS. This proposed clinical trial will provide high-level and valuable clinical evidence that could inform the effect size and personalized modeling of focused low-intensity TUS for different types of brain diseases.

## Trial registration

ClinicalTrials.gov Identifier: NCT06776705.

## Introduction

Chronic tinnitus, as a common symptom, is the persistent perception of sound (e.g., buzzing or ringing) in the absence of external acoustic stimulus, lasting longer than six months [1]. The prevalence of chronic tinnitus rises steadily after age 40 and reaches 24% after age 65 [2,3]. Notable, chronic tinnitus can severely jeopardize sleep quality, brain health, and even lead to cognitive decline in aging populations [4–6]. The status of sleep disturbances is a frequent, but critical comorbidity in chronic tinnitus sufferers with a high prevalence ranging from 50% to 77% [7]. A growing body of data connects poor sleep quality to tinnitus and an increased risk of dementia in old adults [1,7]. Untreated sleep disturbances can cause glymphatic dysfunction and brain atrophy as a result of accelerated cognitive decline in chronic tinnitus sufferers [8–10]. Of note, glymphatic dysfunction can lead to the accumulation of neurotoxic waste products in the brain, potentially contributing to neurological conditions, such as tinnitus and cognitive decline. Poor sleep quality may exacerbate this dysfunction, creating a vicious cycle in which chronic tinnitus disrupt sleep, and sleep disturbances further compromise glymphatic clearance. Thus, targeting this cycle through interventions that address both sleep disturbances and chronic tinnitus could hold significant therapeutic value, particularly in ageing populations.

Understanding the neural underpinnings of the vicious cycle is fundamental to effectively managing co-occurring sleep disturbances and chronic tinnitus. Growing evidence indicates that the disruption of intrinsic neural networks can interfere with the brain activities tagged as default mode network (DMN), adding to sleep disturbances in chronic tinnitus sufferers [11]. DMN is an intrinsic brain network active during resting state and involved in mind-wandering and memory consolidation [12]. Neuroimaging studies found that sleep disturbances are closely related to the dysfunction of DMN. For example, people with sleep deprivation and insomnia showed weaker hippocampal-DMN connectivity [13] and smaller hippocampal volume [14]. It seems that enhancing sleep quality might be a feasible and indirect pathway for breaking down the vicious cycle and reducing the influence of chronic tinnitus on daily functioning. At present, non-pharmacological treatments are very limited for patients suffering from sleep disturbances and chronic tinnitus. With the advantage of targeting neural networks, non-invasive brain stimulation (NIBS), as a safe treatment, holds promise for managing these two conditions simultaneously. For example, clinical trials of transcranial magnetic stimulation (TMS) have been reported to improve the auditory processing, reduce the perception of tinnitus [15,16] and enhance the subjective sleep quality [17–19]. Although there are numerous studies that examine the effects of TMS in treating chronic tinnitus and sleep disturbances, the results are highly varied across the trials. The major reasons of heterogeneity might be summarized into the following three points: (1) Comorbidities: current clinical trials only focus on one symptom (i.e., either tinnitus or sleep disturbances) and provide insufficient evidence to support the use of TMS for managing both sleep disturbances and chronic tinnitus simultaneously. (2) Selection of treatment targets: TMS has very limited power to reach deep brain regions that might be related to sleep process, glymphatic system, and the perception of tinnitus, such as hippocampus. (3) Variability in head size and brain morphometry: the electric fields induced by TMS (i.e., dose) were severely affected by the morphometric features of the target at individual level [20]. Compared to TMS, focused low-intensity transcranial ultrasound stimulation (TUS) is a newly developed modality of NIBS that offers promising therapeutic effects by modulating the glymphatic system and transmitting acoustic energy into brain parenchyma [21,22]. Low-intensity TUS can reach deeper brain regions, such as hippocampus, with a high spatial resolution (i.e., mm) [23], which sparks interest in targeting DMN and managing the comorbidities in ageing populations.

Collectively, there is insufficient research data to support a large, full-scale randomized controlled trial (RCT) that involves investigation of the efficacy and sustainability of focused low-intensity TUS in patients with co-occurring sleep disturbances and chronic tinnitus. There is also a lack of clinical evidence that would allow the estimation of the efficacy of imaging-guided low-intensity TUS or sample size for the full-scale RCT. Thus, this pilot RCT aims to investigate the safety and feasibility of low-intensity hippocampal TUS for sleep disturbances in chronic tinnitus patients that allows to determine the sample size of a full-scale RCT. The findings of this RCT will provide valuable clinical evidence that could inform the effect size and personalized modeling of low-intensity TUS for age-related brain diseases.

## Materials and methods

### Research design

A two-arm RCT will be carried out with repeated measurements, including baseline, 2nd week, 6th week and 12th week. It makes reference to the suggested requirements of a phase II design for non-pharmacological intervention. The study will be conducted following the Recommendations for Interventional Trials (SPIRIT) (Fig 1) and the Consolidated Standards of Reporting Trials (CONSORT) statement (http://www.consort-statement.org) (Fig 2).

### Sample size and power analysis

Until now, no published clinical trials have been conducted in chronic tinnitus patients with sleep disturbances, thus our calculation will be based on published data in similar populations. As this clinical trial is a pilot and feasibility study, the sample size is considered fifteen older patients suffering from sleep disturbances and chronic tinnitus in each study arm. This sample size is appropriate for the primary study aims. To evaluate the potential efficacy of the treatments as compared

| | STUDY PERIOD | | | | |
|---|---|---|---|---|---|
| | **Enrolment** | **Baseline** | **Post-allocation** | | **Close-out** |
| **TIMEPOINT\*\*** | *-t₁* | **0** | *t₁* | *t₂* | *t₃* |
| **ENROLMENT:** | | | | | |
| **Eligibility screen** | X | | | | |
| **Informed consent** | X | | | | |
| **Allocation** | X | | | | |
| **Data collection** | | X | X | X | X |
| **INTERVENTIONS:** | | | | | |
| *Low-intensity TUS* | | ●————● | | | |
| *Sham TUS* | | ●————● | | | |
| **ASSESSMENTS:** | | | | | |
| *Sleep quality* | | X | X | X | X |
| *Tinnitus symptoms* | | X | X | X | X |
| *Glymphatic function* | | X | X | X | X |
| | | | | | |

**Fig 1. Schedule of the pilot randomized clinical trial according to the standard protocol items: Recommendations for interventional trials checklist (SPIRIT).** Abbreviations: TUS = Transcranial ultrasound stimulation.

with the control and assuming a medium standardised effect size (0.5), thirteen participants are required in each group with 80% one-sided CI approach which is suggested for pilot trials [24]. Besides, the sample size in this study is also estimated based on our recent pilot study using transcranial random noise stimulation (tRNS) to treat sleep disturbances (n = 13 for each arm) [25]. Thus, to account for the dropout rate of 10%, the total sample size is calculated as thirty.

## Study population, recruitment and eligibility criteria

Eligible participants will be recruited through our established trial-ready cohort (TRC) i.e., Hong Kong Cohort of Abnormal Sleep in Ageing Population (HK-ASAP) (ClinicalTrials.gov Identifier: NCT06170073) [26]. Potential participants will undergo screening by trained research assistants to assess eligibility and availability for this pilot trial. Both participants and their caregivers will receive a detailed briefing about this trial before providing informed consent.

Potential participants will need to satisfy the following inclusion criteria:

1. Chinese, right-handed, aged from 60 to 80 years.

2. Sleep disturbances are defined as a Pittsburgh Sleep Quality Index (PSQI) total score above five [27].

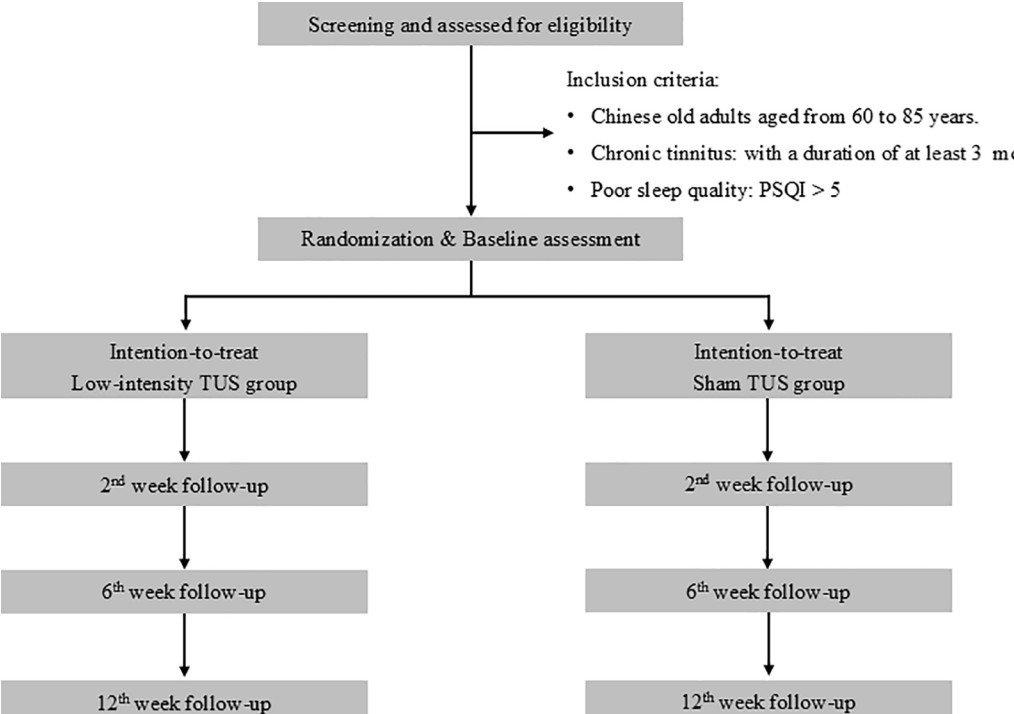

**Fig 2. The Consolidated Standards of Reporting Trials (CONSORT) flow diagram for the clinical trial of focused low-intensity hippocampal transcranial ultrasound stimulation (TUS).**

3. Subjective chronic tinnitus is defined as the tinnitus can only be heard by the patient, and there is no acoustic source [28]. The duration of chronic tinnitus is at least six months (as spontaneous recovery may occur among some patients in the first few months) [29].

4. With no signs of cognitive impairments that are defined as a total score with the Montreal Cognitive Assessment (MoCA) above 26 (adjusted with educational level) [30].

5. No interference with independence in everyday activities.

Exclusion criteria include:

1. Diagnosis of objective tinnitus is defined as the tinnitus produced by an internal acoustic source activating the cochlea and that can be heard by another person.

2. Diseases of ear canal and tympanic membrane checked by otoscopic examination.

3. Diagnosis of Meniere's disease and acoustic neuromas.

4. Past history of neurological or mental disorders.

5. Physically frail affecting attendance to treatment sessions.

6. Already attending regular treatments, such as cognitive behavioral therapy or music therapy.

7. Taking a psychotropic or other medication known to affect hearing functions.

8. Significant communicative impairments, such as severe hearing loss. The diagnosis of deaf or severe hearing loss will be informed by the medical records.

## Ethical issue

The pilot RCT will comply with the Declaration of Helsinki and the Good Clinical Practice guidelines of the International Conference on Harmonisation of technical requirements for registration of pharmaceuticals for human use (ICH-GCP). Ethical standards will be strictly adhered to this study, ensuring that written informed consent will be obtained from the participants and their anonymity, privacy, and confidentiality are respected. Participants will be able to withdraw from the study at any time, with no consequences and without giving any reason. This study has been approved by the Clinical Research Ethics Committee of The Chinese University of Hong Kong (CUHK) and New Territories East Cluster (NTEC) (The Joint CUHK-NTEC) (Date: January 24, 2023, Number: 2023.636) and was registered in the United States National Institute of Health Registration System with Clinical Trials (ClinicalTrials.gov Identifier: NCT06776705). All participant data, including audiometric results and health surveys, were anonymized at collection and stored on password-protected computers. Access was limited to study investigators, and data retention follows IRB guidelines (5 years post-study).

## Pre-treatment neuroimaging

**Structural magnetic resonance imaging.** High-resolution structural magnetic resonance imaging (MRI) scans will be collected at the Prince of Wales Hospital using a 3.0 Tesla Siemens MAGNETOM Prisma MRI scanner (Siemens Healthcare Sector, Erlangen, Germany) equipped with a 32-channel head coil. T1-weighted magnetization prepared rapid gradient echo (MPRAGE) sequence is used to optimize the grey-white contrast, with following parameters [31]: repetition time (TR) = 2070 ms, echo time (TE) = 3.93 ms, axial acquisition with a 256 × 256 × 192 matrix, thickness = 1 mm, no gap, field of view (FOV) = 230 mm, flip angle = 15°. The sequence yields high quality isotropic images with the voxel size of 1 mm × 1 mm × 1 mm.

**Diffusion magnetic resonance imaging.** To evaluate the glymphatic function, diffusion-weighted imaging (DWI) will be acquired in 30 independent directions along with five interleaved non-diffusion weighted (b = 0) images (1.9 mm3 resolution, TR = 8500 ms, TE = 81 ms, b = 700, echo spacing = 0.69 ms, GRAPPA iPAT factor = 2, 72 slices, 243 × 243 × 137 mm FOV). An accompanying phase map image is acquired using the same shim as the DWI sequence to correct for field inhomogeneities (4 mm$^3$ resolution, TR = 1000 ms, TE = 3.60/6.06 ms, FA = 90°, 48 slices, 256 × 256 × 230 mm FOV).

## Randomization, allocation and masking

To ensure the homogeneous distribution between groups, all the eligible participants will be randomly assigned to receive either low-intensity TUS or sham TUS following a 1:1 ratio. A randomization assignment will be generated prior to study enrollment using an online system (http://randomization.com/) by a statistician who is not involved in the study design. After the randomization and baseline assessment, the participants will be scheduled for treatment sessions. Assessment staff and participants will be blinded to the study design and group allocation. The researcher who administers the TUS treatment will not participate in the assessments.

## Treatment strategies

**Apparatus and settings.** NeuroFUS TPO and CTX-500-4 transducer (Brainbox Ltd., Cardiff, UK) are used in this clinical trial. This TUS system consists of a four-element ultrasound transducer (64 mm diameter) with a central frequency of 500 kHz [23]. We will use the theta-burst low-intensity TUS protocol with the following parameters [32]: pulse

duration = 20 milliseconds, pulse repetition interval = 200 milliseconds and total duration = 80 seconds, giving a total of 400 pulses.

Based on individual's structural MRI, we will construct head model of TUS and perform the transcranial acoustic simulations to ensure that we remain below the FDA guidelines for diagnostic ultrasound (MI ≤ 1.9; ISPPA ≤ 190 W/cm2) after transcranial transmission. We prepare each participant's head by parting any hair over the intended target and applying ultrasound transmission gel (Aquasonic 100, Parker Laboratories Inc.).

### Stimulation modalities

**Low-intensity transcranial ultrasound stimulation (TUS).** Theta-burst TUS (tbTUS) protocol will be used in this study for modulating neuronal activities and enhancing the neuroplasticity in hippocampal circuits [32,33]. The stimulation parameters of low-intensity TUS include Frequency: 5 Hz, pulse duration = 20 milliseconds, pulse repetition interval = 200 milliseconds and the total duration = 80 seconds, giving a total of 400 pulses.

**Sham TUS.** For sham TUS (Placebo-controlled group), the transducer with ultrasound gel will be placed at the same location as active condition for 80 seconds with TPO power-off. This procedure mimics the skin sensation induced by ultrasound gel without producing any effects. To control auditory effects, we played a sound mimicking the pulse repetition and duration of low-intensity TUS.

### Grouping and treatment schedule

According to the design of this study, all the eligible participants will be randomly assigned to two groups: (1) Low-intensity TUS; (2) Sham TUS. After baseline assessment, the participants will start to receive a total of six successive sessions of low-intensity TUS treatment. This pilot trial is a 2-week treatment with three sessions per week, lasting for two weeks. Each session of treatment lasts for 80 seconds. The schedule for treatment is the same in the two randomized groups.

### Safety assessments

**Adverse events checklist.** A checklist of adverse events associated with low-intensity TUS is applied to monitor tolerability and potential side effects. The Adverse Event Checklist (AEC) covering the symptoms from eight systems, including body as a whole, nervous system, cardiovascular, circulatory, digestive, musculoskeletal, urogenital systems and metabolic, is conducted at each assessment points through this pilot RCT [34].

### Outcome measures

**Primary outcomes.**

1. Subjective sleep quality: The Pittsburgh Sleep Quality Index (PSQI), as a 19-item self-report questionnaire, is used to evaluate the subjective sleep quality in a month [27,35]. The PSQI contains seven components. The subscore of each component ranges from 0 to 3, and the maximum total composite score of the PSQI is 21. The sum of these component scores yields a measure of global sleep quality. The cutoff score of poor sleep quality is 5 or more.

2. Severity of tinnitus: Tinnitus functional index (TFI) is a 25-item self-administered questionnaire which assesses eight domains of tinnitus impact, including intrusiveness; sense of control; cognitive; sleep; auditory; relaxation; quality of life (QoL); emotion [36]. The TFI Chinese version (TFI-CH) has been validated for clinical and research purposes in Hong Kong [37].

**Secondary outcomes.**

1. Objective sleep quality: Actigraphic records were used to quantify sleep-wake cycle and estimate the objective sleep efficiency. The actigraph is about the size of a wristwatch and is usually worn on the wrist continuously for multiple days and nights, which can be used as an objective measurement of sleep quality and sleep-wake cycle in older adults.

2. The Tinnitus Handicap Inventory (THI), as a reliable 25-item self-administered tool, is widely used in the studies of tinnitus treatment efficacy. The THI Chinese version (THI-CH) has been validated for clinical and research purposes in Hong Kong [38].

3. Glymphatic function is assessed by the index of Diffusion Tensor Image Analysis ALong the Perivascular Space (DTI-ALPS) through DWI data [10]. DTI-ALPS is the quantification of the directional diffusion of water molecules with the perivascular space (PVS). The diffusivities in the directions of the x-axis ($D_x$), y-axis ($D_y$) and z-axis ($D_z$) of the regions of interest (ROIs) on projection fibres and association fibres are recorded as $D_{xproj}$, $D_{yproj}$, $D_{zproj}$, $D_{xassoc}$, $D_{yassoc}$, $D_{zassoc}$, respectively [10,39].

## Assessment schedule

The comprehensive assessments, including sleep quality, severity of tinnitus, and adverse events, will be conducted at 1 week before low-intensity TUS treatment (baseline, T0), and at 2nd week (T1), 6th week (T2), and 12th week (T3). Actigraphic records and Glymphatic function will be assessed at baseline and 2nd week.

## Statistical analysis

The data analyst will be blinded to the randomization of participants. Analyses will be primarily conducted by the intention-to-treat (ITT) approach. The measurement scales for primary and secondary outcomes are continuous. Linear mixed effect models will be used to evaluate the differences between the conditions on primary and secondary outcomes measured at each time point. This statistical method will facilitate the inclusion of the participants with missing data. For handling missing data, multiple imputation will be used in this study. Treatment, time points, and their interaction will be modelled as fixed effects. Participants will be modelled as random effects at each time point. Pre-treatment subjective sleep quality, tinnitus symptoms, and cognitive performance will be compared between the two randomized groups. Score changes of sleep quality and tinnitus symptoms from baseline to follow-up points across randomized groups will be tested with occasions (time points) at level one and participants at level two. Covariates, including age, duration of tinnitus and PSQI total score, identified from baseline differences will be entered in the regression model. Secondary analyses of groupwise differences in the changes of glymphatic function, hearing function, and their associations between changes of tinnitus symptoms and sleep quality will be performed. We will also monitor the incidence of adverse events and the characteristics of program adherence. Statistical significance will be set at 2-sided $p < 0.05$. Computations will be performed using R Studio (version 1.1.456).

## Study timeline

The trial has been scheduled to take place from the 1st of January 2025 to the 1st of June 2026. By the end of February 2025, we successfully screened fourteen eligible participants who have completed MRI scanning.

## Discussion

This study presents the rationale and research protocol for a double-blind, sham-controlled pilot randomized clinical trial of using low-intensity transcranial ultrasound stimulation (TUS) to treat sleep disturbances in patients with chronic tinnitus. With the advantage of stimulating deep brain structures, low-intensity TUS has great potential for effectively modulating

the activities of DMN, particularly hippocampus. Based on the hypothesis of targeting the "vicious cycle", we expected that the patients who receive focused low-intensity TUS treatment will have more alleviations on sleep disturbances and tinnitus symptoms. It is also expected that the enhanced sleep quality and tinnitus symptoms will be related to better glymphatic function. This pilot trial stands out for a neuroscience-driven approach to address the non-pharmacological and non-invasive management of the comorbidities in older adults.

To our knowledge, this is the first randomized clinical trial evaluating the safety, feasibility and efficacy of focused low-intensity hippocampal TUS for treating sleep disturbances in chronic tinnitus patients. Adhering to the CONSORT checklist, this pilot RCT will employ rigorous methodology, including randomization by a blinded assessor and conceal-ment of treatment differences from participants. Participants will be randomized by a blinded assessor and will not receive grouping information. Assessors and statistician will remain blinded to group allocation and have no contact with partic-ipants during the treatment period. The findings will provide high-level clinical evidence on the therapeutic potential of focused low-intensity hippocampal TUS for sleep disturbances in the clinical populations.

The findings of this pilot RCT will be circulated to international peer-reviewed journals, symposium and scientific confer-ences, regardless of whether the results are positive, negative or inconclusive in relation to the hypothesis.

## Limitations

Although pilot RCT is essential for a full-size clinical trial, there are several key limitations that should be carefully consid-ered when interpretating the results. First, pilot RCT involves a small sample size of participants which may reduce the statistical power. Second, the primary aim of pilot RCT is to test the feasibility of a novel treatment that has underpower to detect small but clinically meaningful effects. Third, this pilot RCT is an offline TUS treatment. Although pre-treatment simulation will be used for locating the target, no online EEG or fMRI will be applied for verifying whether TUS actually modulates the hippocampus. Fourth, objective assessment of hearing function was not included in this study. Fifth, the sample size calculation was based on a medium effect size (0.5) without prior TUS data. Due to the limited sample size, the results may not be generalizable to larger populations. Their limitations highlight the need for cautious interpretation and proper planning for subsequent full-scale clinical trials.

## Conclusion

This pilot RCT will yield new and valuable evidence on the effects of low-intensity hippocampal transcranial ultrasound stimulation on sleep quality, glymphatic function, and tinnitus symptoms. The findings will enhance our understanding of the interrelationships between these factors and their relevance to hippocampal functions and brain health. Ultimately, this study may pave the way for developing innovative, full-scale therapeutic interventions targeting age-related brain disor-ders and their comorbidities in ageing populations.

## Supporting information

**S1 File. SPIRIT checklist.**
(PDF)

**S2 File. Research protocol.**
(PDF)

## Author contributions

**Conceptualization:** Hanna Lu.

**Investigation:** Xi Ni, Yuk Shan Yuen, Kaipeng Wang, Natalie Shu Yang.

**Methodology:** Xi Ni, Zeyan Li, Liwei Guo, Hanna Lu.

**Project administration:** Hanna Lu.

**Resources:** Hanna Lu.

**Software:** Zeyan Li, Liwei Guo.

**Writing – original draft:** Hanna Lu.

**Writing – review & editing:** Xi Ni, Yi Yuan, Lin Meng, Hanna Lu.

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
