## [Decision Letter · Decision Letter 0]

13 May 2025

PONE-D-25-13152Focused low-intensity transcranial ultrasound stimulation (TUS) for sleep disturbances in patients with chronic tinnitus: A study protocol for a pilot randomized controlled trialPLOS ONE

Dear Dr. Lu,

Thank you for submitting your manuscript to PLOS ONE. After careful consideration, we feel that it has merit but does not fully meet PLOS ONE’s publication criteria as it currently stands. Therefore, we invite you to submit a revised version of the manuscript that addresses the points raised during the review process.

We look forward to receiving your revised manuscript.

Kind regards,

Hantong Hu

Academic Editor

PLOS ONE

- https://doi.org/10.1371/journal.pone.0289591

In your revision ensure you cite all your sources (including your own works), and quote or rephrase any duplicated text outside the methods section. Further consideration is dependent on these concerns being addressed.

4. Please include a separate caption for each figure in your manuscript

Additional Editor Comments (if provided):

Reviewers' comments:

Reviewer's Responses to Questions

**Comments to the Author**

1. Does the manuscript provide a valid rationale for the proposed study, with clearly identified and justified research questions?

Reviewer #1: Yes

Reviewer #2: Yes

Reviewer #3: Yes

2. Is the protocol technically sound and planned in a manner that will lead to a meaningful outcome and allow testing the stated hypotheses?

Reviewer #1: Yes

Reviewer #2: Yes

Reviewer #3: Yes

3. Is the methodology feasible and described in sufficient detail to allow the work to be replicable?

Reviewer #1: Yes

Reviewer #2: Yes

Reviewer #3: No

4. Have the authors described where all data underlying the findings will be made available when the study is complete?

Reviewer #1: Yes

Reviewer #2: Yes

Reviewer #3: Yes

5. Is the manuscript presented in an intelligible fashion and written in standard English?

Reviewer #1: Yes

Reviewer #2: Yes

Reviewer #3: Yes

6. Review Comments to the Author

You may also provide optional suggestions and comments to authors that they might find helpful in planning their study.

Reviewer #1: Dear Editor,

Many thanks for the opportunity to review the manuscript titled “Focused low-intensity transcranial ultrasound stimulation (TUS) for sleep disturbances in patients with chronic tinnitus: A study protocol for a pilot randomized controlled trial”

Abstract:

Are you referring to older or senior patients for the target population? What are you referring to as senior patients? Kindly clarify

Be specific on the random assignment procedure, is it 1:1, 1:2 and type of randomization (single, multiple arm etc)

What is Sham TUS? Let’s use standard nomenclature. Do you mean control? The word sham is a bit problematic

Study Design

Authors have provided details on the study. A two-arm RCT with repeated measurements, including baseline, 2nd week, 6th week and 12th week is a appropriate. However, no justification was provided for implementing the intervention for 12 weeks. Why not 10 or 15 weeks or any other week. Some justification must be provided because the duration of any intervention has significant effect on the outcome measures

Once again, kindly check the use of the word senior. Older adults aged 60-70 years is more appropriate.

Sample size determination is quite detail and is ok.

Statistical Methods

The measurement scale (continuous, binary, multinomial, ordinal, etc.) for most primary and secondary outcome measures was unclear. This will inform the kind of statistical models that will be required to quantify the effect of the intervention.

Replace linear mixed models with linear mixed effect models. The choice of employing linear mixed Effect models is appropriate, but the measurement scale should be clearly defined to determine whether these models will be appropriate for all the outcome measures. This is important because the authors indicated that the cutoff score for poor sleep quality is 5 or more. If this outcome is treated as binary, then ideally, a mixed-effect logistic regression model will be more appropriate.

Authors did not mention key assumptions under linear mixed effect models and how they will address issues related to the deviation from the key assumption.

Methods to address potential missing observations should be included in the methods section

Reviewer #2: Thank for the opportunity to review this protocol and pilot randomized control trial. Xi and colleagues propose to investigate the safety and feasibility of applying low-intensity hippocampal TUS for sleep disturbances in chronic tinnitus with the goal of determining a sample size for a future full-scale RCT. The proposed study addresses a clinical gap with the potential for significant impact. The pilot design is appropriate for evaluating feasibility and informing a subsequent full-scale trial. There is a clear recruitment strategy and analysis plan. Overall, my comments are minor and focused on improving the contextualization and rationale for the study, particularly in the introduction. Additionally, the authors may consider clearly defining adverse events and discussing limitations specific to the pilot design explicitly.

Introduction

The authors may consider including a brief definition for tinnitus and some common etiologies/ pathophysiology for broader contextualization.

Some epidemiology of chronic tinnitus may also help readers to understand the scope of your research question.

Additionally, while the authors do establish a relationship between hearing loss and tinnitus, the nature of this association becomes unclear throughout the paragraph. Consider clarifying the interplay between these conditions.

“Growing evidence indicates that the disruption of intrinsic neural networks can interfere with the brain activities tagged as default mode network (DMN), adding to sleep disturbances in tinnitus sufferers” I would suggest providing some more context for this here—why is this important in the context of your study? This is somewhat addressed later on but consider bringing more of that discussion here.

Additionally, consider reworking paragraph 2 for improved flow as currently it jumps around quite a bit.

“current clinical trial is lacking for TMS that are effective in the management of sleep disturbances and chronic tinnitus simultaneously” Do the authors mean that current TMS trials do not include patients with co-existing tinnitus and sleep disturbance? Or are you citing a specific trial here? Please clarify

Has TUS been applied successfully to any other pathologies?

Methods

Research Design

Please expand the research design parameters in the text. What measurements will be taken at each timepoint? At which timepoint is TUS delivered? This is nicely summarized in the figure but please include a description in the text as well.

Figure 1 is very difficult to read. Text is fuzzy.

Sample Size and Power Analysis

“To evaluate the potential efficacy of the treatments as compared with the control and assuming a medium standardised effect size (0.5), thirteen participants are required in each group with 80% on-sided CI approach which is suggested for pilot trials.” I believe the authors meant “one-sided.”

Study Population, Recruitment, and Eligibility Criteria

Severe hearing loss is listed as exclusion criteria. How is severe hearing loss defined for your study? Is this based on previous diagnosis, or will patients be screened for hearing loss beforehand? Is there a threshold hearing loss that will not be eligible?

Ethics

How will your data be handled and stored securely?

What qualifies as an adverse event in your study? Who is responsible for monitoring such events and how will they be reported?

Imaging

Clarify within the text at which timepoints in your study design these images (MRI and DWI) will be taken, and for what purpose.

In the section on DWI, expand on DTI-ALPS method.

Simulation Modality

How were these intensity parameters decided upon? Are they based on animal studies?

Discussion

Consider including the rationale for hippocampal brain region directed TUS. Is this based on previous research in TMS or TUS? Underlying pathophysiology of tinnitus or sleep disturbance?

Please include a limitations section.

Figure 1

Consider including a key to define each timepoint.

Reviewer #3: The manuscript is well-structured and presents a compelling study protocol for a pilot RCT investigating low-intensity transcranial ultrasound stimulation (TUS) for sleep disturbances in chronic tinnitus patients. Below is a detailed review with suggestions for improvement.

Suggestions:

•Abstract:

Specify the primary/secondary outcomes more succinctly (e.g., "Primary: PSQI; Secondary: actigraphy, THI, DTI-ALPS").

Clarify the intervention duration (e.g., "2-week treatment, 3 sessions/week, 80s/session").

•Introduction:

Streamline the "vicious cycle" explanation (glymphatic dysfunction → tinnitus → sleep disruption) to avoid redundancy.

Define DMN (Default Mode Network

Clear justification for TUS over TMS (e.g., deeper targeting of hippocampus, glymphatic modulation).

•Methods:

Randomization: Clarify how blinding is maintained (e.g., who administers TUS vs. assesses outcomes?).

Sham Protocol: Reiterate how auditory masking ensures blinding (currently mentioned but could be emphasized).

•TUS Parameters:

Justify the choice of theta-burst protocol (cite preclinical/clinical evidence for hippocampus modulation).

Add safety details (e.g., how often will MRI/thermal checks be performed during treatment?).

•Glymphatic Measurement: Briefly explain DTI-ALPS and its validation in tinnitus/sleep studies (or cite references).

•How will you verify TUS actually modulates the hippocampus? (fMRI/EEG planned?)

•Why choose a 2-week intervention? Is this based on prior TUS durability data?

•How will you handle missing data (e.g., multiple imputation)?

•Statistical Analysis

Power Analysis: Acknowledge limitations of assuming a medium effect size (0.5) without prior TUS data. Consider a sensitivity analysis.

Covariates: List potential covariates (e.g., baseline PSQI, tinnitus duration, age) to be included in mixed models.

7. PLOS authors have the option to publish the peer review history of their article (what does this mean? ). If published, this will include your full peer review and any attached files.

**Do you want your identity to be public for this peer review?** For information about this choice, including consent withdrawal, please see our Privacy Policy .

Reviewer #1: No

Reviewer #2: No

Reviewer #3: No

---

## [Author Response · Author response to Decision Letter 1]

14 Jun 2025

Point-by-point responses to the reviewer’s comments on the manuscript

“Focused low-intensity hippocampal transcranial ultrasound stimulation (TUS) for sleep disturbances in patients with chronic tinnitus: A study protocol for a pilot randomized controlled trial”

Reviewer #1:

Many thanks for the opportunity to review the manuscript titled “Focused low-intensity transcranial ultrasound stimulation (TUS) for sleep disturbances in patients with chronic tinnitus: A study protocol for a pilot randomized controlled trial”

Comment 1: Abstract:

Are you referring to older or senior patients for the target population? What are you referring to as senior patients? Kindly clarify. Be specific on the random assignment procedure, is it 1:1, 1:2 and type of randomization (single, multiple arm etc). What is Sham TUS? Let’s use standard nomenclature. Do you mean control? The word sham is a bit problematic.

Response 1:

Thank you so much for reviewing our paper and sharing your valuable suggestions with us. Yes. This study is focusing on older patients with an age range of 60 to 90 years. As you suggested, we have modified the description of sham TUS. Since the word “sham” is a common description of placebo condition in the field of neuromodution, thus, we keep both of the description for clearly defining the placebo condition.

Comment 2: Study Design

Authors have provided details on the study. A two-arm RCT with repeated measurements, including baseline, 2nd week, 6th week and 12th week is a appropriate. However, no justification was provided for implementing the intervention for 12 weeks. Why not 10 or 15 weeks or any other week. Some justification must be provided because the duration of any intervention has significant effect on the outcome measures. Once again, kindly check the use of the word senior. Older adults aged 60-70 years is more appropriate.

Response 2:

Thank you for your insightful comment. All the senior patients have been changed as “older patients”. The rationale why we conduct the follow-up assessment at 2nd, 6th and 12th week contains two aspects: 1) The treatment will last for 2 weeks, thus, we need to arrange the 1st time post-treatment assessment within a week. 2) One of our key primary outcomes is subjective sleep quality measured by PSQI, which is a questionnaire that assesses sleep quality and disturbances over a one-month period.

Comment 3: Statistical Methods

The measurement scale (continuous, binary, multinomial, ordinal, etc.) for most primary and secondary outcome measures was unclear. This will inform the kind of statistical models that will be required to quantify the effect of the intervention.

Replace linear mixed models with linear mixed effect models. The choice of employing linear mixed Effect models is appropriate, but the measurement scale should be clearly defined to determine whether these models will be appropriate for all the outcome measures. This is important because the authors indicated that the cutoff score for poor sleep quality is 5 or more. If this outcome is treated as binary, then ideally, a mixed-effect logistic regression model will be more appropriate.

Methods to address potential missing observations should be included in the methods section

Response 3: Thank you for your helpful suggestion! The measurement scale has been added in the statistical analysis. For handling missing data, multiple imputation will be used in this study. Considering sleep quality with continuous scale will be used as outcome, we will use linear mixed effect models as you suggested. The modified part in statistical analysis is on Page 13 as “The measurement scales for primary and secondary outcomes are continuous. Linear mixed effect models will be used to evaluate the differences between the conditions on primary and secondary outcomes measured at each time point. This statistical method will facilitate the inclusion of the participants with missing data. For handling missing data, multiple imputation will be used in this study.”.

Reviewer #2:

General comment: Thank for the opportunity to review this protocol and pilot randomized control trial. Xi and colleagues propose to investigate the safety and feasibility of applying low-intensity hippocampal TUS for sleep disturbances in chronic tinnitus with the goal of determining a sample size for a future full-scale RCT. The proposed study addresses a clinical gap with the potential for significant impact. The pilot design is appropriate for evaluating feasibility and informing a subsequent full-scale trial. There is a clear recruitment strategy and analysis plan. Overall, my comments are minor and focused on improving the contextualization and rationale for the study, particularly in the introduction. Additionally, the authors may consider clearly defining adverse events and discussing limitations specific to the pilot design explicitly.

Response:

Thank you so much for reviewing our paper and sharing your valuable suggestions with us. A checklist of adverse events associated with low-intensity TUS is applied to monitor tolerability and potential side effects. The Adverse Event Checklist (AEC) covering the symptoms from eight systems, including body as a whole, nervous system, cardiovascular, circulatory, digestive, musculoskeletal, urogenital systems and metabolic, is conducted at each assessment points through this pilot RCT. The updated contents were on page 11.

Comment 1: Introduction

The authors may consider including a brief definition for tinnitus and some common etiologies/ pathophysiology for broader contextualization. Some epidemiology of chronic tinnitus may also help readers to understand the scope of your research question. Additionally, while the authors do establish a relationship between hearing loss and tinnitus, the nature of this association becomes unclear throughout the paragraph. Consider clarifying the interplay between these conditions.

Response 1:

Thank you very much for pointing out this issue. After carefully checking, chronic tinnitus itself does not directly cause hearing loss. Because we do not include a hearing test (audiogram) Thus, we removed the hearing loss in the contents. Meanwhile, we added a brief introduction in the first paragraph and highlighted the “vicious cycle” of sleep-tinnitus-glymphatic system on Page 4.

Comment 2:

“Growing evidence indicates that the disruption of intrinsic neural networks can interfere with the brain activities tagged as default mode network (DMN), adding to sleep disturbances in tinnitus sufferers” I would suggest providing some more context for this here—why is this important in the context of your study? This is somewhat addressed later on but consider bringing more of that discussion here. Additionally, consider reworking paragraph 2 for improved flow as currently it jumps around quite a bit.

Response 2:

Thank you for this advice. We enhanced this part through introducing the role of DMN in sleep process as “DMN is an intrinsic brain network active during resting state and involved in mind-wandering and memory consolidation. Neuroimaging studies found that sleep disturbances are closely related to the dysfunction of DMN. For example, people with sleep deprivation and insomnia showed weaker hippocampal-DMN connectivity and smaller hippocampal volume. It seems that enhancing sleep quality might be a feasible and indirect pathway for breaking down the vicious cycle and reducing the influence of tinnitus on daily functioning.” on Page 4-5. The newly added references 12-14 have also been updated in the reference list on Page 18.

Comment 3:

“current clinical trial is lacking for TMS that are effective in the management of sleep disturbances and chronic tinnitus simultaneously” Do the authors mean that current TMS trials do not include patients with co-existing tinnitus and sleep disturbance? Or are you citing a specific trial here? Please clarify

Response 3:

Yes. You are right. Previous and current TMS studies focus on single symptom, i.e., either tinnitus or sleep disturbances. In this revision, we listed some results of clinical trials to clarify the point of view. The newly added references 15-19 have been updated in the reference list on Page 18-19.

Comment 4: Methods-Research Design

Please expand the research design parameters in the text. What measurements will be taken at each timepoint? At which timepoint is TUS delivered? This is nicely summarized in the figure but please include a description in the text as well. Figure 1 is very difficult to read. Text is fuzzy. Consider including a key to define each timepoint.

Response 4:

Thank you for the suggestion. We have clarified the assessment schedule and measurements of assessments as “The comprehensive assessments, including sleep quality, severity of tinnitus, and adverse events, will be conducted at 1 week before low-intensity TUS treatment (baseline, T0), and at 2nd week (T1), 6th week (T2), and 12th week (T3). Actigraphic records and Glymphatic function will be assessed at baseline and 2nd week.” on Page 13. The layout of figure 1 is based on the template of SPIRIT (journal’s template). I am sorry that figure 1 is same as the previous one.

Comment 5: Sample Size and Power Analysis

“To evaluate the potential efficacy of the treatments as compared with the control and assuming a medium standardised effect size (0.5), thirteen participants are required in each group with 80% on-sided CI approach which is suggested for pilot trials.” I believe the authors meant “one-sided.”

Response 5:

Yes. You are right. The typo has been corrected. Thank you!

Comment 6:

Study Population, Recruitment, and Eligibility Criteria

Severe hearing loss is listed as exclusion criteria. How is severe hearing loss defined for your study? Is this based on previous diagnosis, or will patients be screened for hearing loss beforehand? Is there a threshold hearing loss that will not be eligible?

Response 6:

Thank you for your insightful question. We listed severe hearing loss as exclusion criteria based on (1) the medical history and (2) the score of Hearing Handicap Inventory for Adults (HHIA). In the screening stage, we collected the data of HHIA, which is a self-evaluation of hearing loss.

Comment 7:

Ethics: How will your data be handled and stored securely?

Response 7:

The security of clinical trial data is a top priority. In this project, identifiable information, such as ID and age on the MRI images, will be removed entirely to protect participant privacy. The hard copies of the questionnaire will be stored in a locked room. The inputted data will be encrypted during collection and transmission to prevent unauthorized access.

Comment 8:

What qualifies as an adverse event in your study? Who is responsible for monitoring such events and how will they be reported?

Response 8:

A self-reported checklist of adverse events associated with low-intensity TUS is applied to monitor tolerability and potential side effects. The Adverse Event Checklist (AEC) covering the symptoms from eight systems, including body as a whole, nervous system, cardiovascular, circulatory, digestive, musculoskeletal, urogenital systems and metabolic, is conducted at each assessment points through this pilot RCT. The PI (Hanna Lu) and the post-doctoral scientist (Xi Ni) are responsible for monitoring such events.

Comment 9:

Imaging: Clarify within the text at which timepoints in your study design these images (MRI and DWI) will be taken, and for what purpose. In the section on DWI, expand on DTI-ALPS method.

Response 9:

DTI-ALPS is the quantification of the directional diffusion of water molecules with the perivascular space (PVS). The diffusivities in the directions of the x-axis (Dx), y-axis (Dy) and z-axis (Dz) of the regions of interest (ROIs) on projection fibres and association fibres are recorded as Dxproj, Dyproj, Dzproj, Dxassoc, Dyassoc, Dzassoc, respectively.

Comment 10: Simulation Modality

How were these intensity parameters decided upon? Are they based on animal studies?

Response 10:

The intensity parameters were from the recent human studies, such as reference [32] Yaakub SN, White TA, Roberts J, Martin E, Verhagen L, Stagg CJ, Fouragnan EF. Transcranial focused ultrasound-mediated neurochemical and functional connectivity changes in deep cortical regions in humans. Nature Communications. 2023; 14(1): 5318.

Comment 11: Discussion

Consider including the rationale for hippocampal brain region directed TUS. Is this based on previous research in TMS or TUS? Underlying pathophysiology of tinnitus or sleep disturbance?

Response 11:

Thank you for your comment. Yes, the rationale is based on the previous studies of TMS. We have included the contents as “With the advantage of stimulating deep brain structures, low-intensity TUS has great potential for effectively modulating the activities of DMN, particularly hippocampus.” on Page 14.

Comment 12: Please include a limitations section.

Response 12:

The limitation section has been updated on Page 15. Although pilot RCT is essential for a full-size clinical trial, there are several key limitations that should be carefully considered when interpretating the results. First, pilot RCT involves a small sample size of participants which may reduce the statistical power. Second, the primary aim of pilot RCT is to test the feasibility of a novel treatment that has underpower to detect small but clinically meaningful effects. Third, this pilot RCT is an offline TUS treatment. Although pre-treatment simulation will be used for locating the target, no online EEG or fMRI will be applied for verifying whether TUS actually modulates the hippocampus. Fourth, objective assessment of hearing function was not included in this study. Fifth, the sample size calculation was based on a medium effect size (0.5) without prior TUS data. Due to the limited sample size, the results may not be generalizable to larger populations. Their limitations highlight the need for cautious interpretation and proper planning for subsequent full-scale clinical trials.

Reviewer #3:

General comment: The manuscript is well-structured and presents a compelling study protocol for a pilot RCT investigating low-intensity transcranial ultrasound stimulation (TUS) for sleep disturbances in chronic tinnitus patients. Below is a detailed review with suggestions for improvement.

Suggestion 1: Abstract:

(1) Specify the primary/secondary outcomes more succinctly (e.g., "Primary: PSQI; Secondary: actigraphy, THI, DTI-ALPS"); (2) Clarify the intervention duration (e.g., "2-week treatment, 3 sessions/week, 80s/session").

Response 1:

Thank you so much for reviewing our paper and sharing your valuable suggestions with us! As you suggested, the outcome measurements have been added in the abstract on Page 3.

Suggestion 2: Introduction: Streamline the "vicious cycle" explanation (glymphatic dysfunction → tinnitus → sleep disruption) to avoid redundancy.

Response 2:

Thank you for your suggestion. We have deleted some words for avoiding redundancy. However, we believe that a very short description of glymphatic function is necessary for explaining the “vicious cycle”. Thus, we kept this part in the main contents.

Suggestion 3: Define DMN (Default Mode Network)

Response 3:

The definition and role of DMN has been updated on Page 4. DMN is an intrinsic brain network active during resting state and involved in mind-wandering and memory consolidation. Neuroimaging studies found that sleep disturbances are closely related to the dysfunction of DMN. For example, people with sleep deprivation and insomnia showed weaker hippocampal-DMN connectivity and smaller hippocampal volume. It seems that enhancing sleep quality might be a feasible and indirect pathway for breaking down the vicious cycle and reducing the influence of tinnitus on daily functioning.

Suggestion 4:

Clear justification for TUS over TMS (e.g., deeper targeting of hippocampus, glymphatic modulation).

Response 4:

Yes. The contents have been modified on page 5, as “Compared to TMS, focused low-intensity transcranial ultrasound stimulation (TUS) is a newly developed modality of NIBS that offers promising therapeutic effects by modulating the glymphatic system and transmitting acoustic en

---

## [Decision Letter · Decision Letter 1]

15 Jul 2025

PONE-D-25-13152R1Focused low-intensity hippocampal transcranial ultrasound stimulation (TUS) for sleep disturbances in patients with chronic tinnitus: A study protocol for a pilot randomized controlled trialPLOS ONE

Dear Dr. Lu,

Thank you for submitting your manuscript to PLOS ONE. After careful consideration, we feel that it has merit but does not fully meet PLOS ONE’s publication criteria as it currently stands. Therefore, we invite you to submit a revised version of the manuscript that addresses the points raised during the review process.

We look forward to receiving your revised manuscript.

Kind regards,

Hantong Hu

Academic Editor

PLOS ONE

Journal Requirements:

Reviewers' comments:

Reviewer's Responses to Questions

**Comments to the Author**

1. Does the manuscript provide a valid rationale for the proposed study, with clearly identified and justified research questions?

Reviewer #1: Yes

Reviewer #2: Yes

Reviewer #3: Yes

2. Is the protocol technically sound and planned in a manner that will lead to a meaningful outcome and allow testing the stated hypotheses?

Reviewer #1: Yes

Reviewer #2: Yes

Reviewer #3: Yes

3. Is the methodology feasible and described in sufficient detail to allow the work to be replicable?

Reviewer #1: Yes

Reviewer #2: Yes

Reviewer #3: Yes

4. Have the authors described where all data underlying the findings will be made available when the study is complete?

Reviewer #1: Yes

Reviewer #2: Yes

Reviewer #3: Yes

5. Is the manuscript presented in an intelligible fashion and written in standard English?

Reviewer #1: Yes

Reviewer #2: Yes

Reviewer #3: Yes

6. Review Comments to the Author

You may also provide optional suggestions and comments to authors that they might find helpful in planning their study.

Reviewer #1: Authors have addressed all my previous comments and also provided clarity when the need arises. The manuscript has improved significantly

Reviewer #2: Thank you for the opportunity to review the revised version of this manuscript. The authors have addressed many of the concerns raised in the initial review, and the paper has improved in clarity and rigor.

Below I outline a few remaining issues and suggestions for further strengthening the manuscript.

Introduction

The authors have effectively revised the introduction to include more background details and epidemiology on tinnitus. However, various sentences in the discussion of the “vicious cycle” remain disjointed and unclear.

For example, “Untreated sleep disturbances can cause glymphatic dysfunction and brain atrophy as a result of accelerated cognitive decline in chronic tinnitus sufferers”

The structure of this sentence makes the cause effect relationship unclear implying that cognitive decline causes glymphatic dysfuntion and brain atrophy rather than sleep disturbance—> glymphatic dysfunction-> brain atrophy-> cognitive decline. Please rework this section to clarify. Please also make clear which aspects are supported in the literature versus which aspects of this relationship are hypothesis.

The addition of more details about the DMN is effective and hepls to clarify the study aims.

“Comorbidities: current clinical trials are lacking for the evidence of using TMS to manage sleep disturbances and chronic tinnitus simultaneously.” While the authors have added the related studies to the references, this sentence remains unclear. Please reword to clarify that current trials focus exclusively on one symptom, either tinnitus or sleep disturbance.

Methods

Please include your justification and parameters for excluding severe hearing loss in the text.

Please include details about data handling and security in the ethics section of the text.

The authors have included a reference to justify their intensity parameters; however, a brief contextual explanation of this reference is lacking in the text. Please elaborate on how this reference supports your point.

The addition of a subheading for safety and adverse events and the subsequent discussion of the adverse events checklist in the methods section is appreciated.

Discussion

Enhanced discussion of targeting deeper brain structures with TUS is effective.

Thank you for including a comprehensive limitations section. The authors now acknowledge limitations more clearly, which is appreciated.

Finally, the manuscript would benefit from thorough language editing, as several sections contain grammatical errors and unclear or informal phrasing that obscure the intended meaning. Improving the English will help ensure the scientific points are more effectively conveyed.

The following are some language points that can be improved.

Till: use until (abstract)

“Toward effectively managing the co-occurring sleep disturbances and chronic tinnitus, it is fundamental for understanding the neural underpinnings of the vicious cycle” (intro)

Suggest rephrasing for clarity (i.e. Understanding the neural underpinnings of the vicious cycle is fundamental to effectively managing co-occuring sleep disturbances and chronic tinnitus.)

“There are a bunch of studies” (intro) This phrase is too informal for scientific writing; consider replacing with “numerous studies have shown…”

Until to date: Revise to “To date,” or “until the present,” or “until now” (methods)

Reviewer #3: Comments have been addressed and answered completely in prefect way.

The manuscript is well-structured and methodology well done

No fruther comments

7. PLOS authors have the option to publish the peer review history of their article (what does this mean? ). If published, this will include your full peer review and any attached files.

**Do you want your identity to be public for this peer review?** For information about this choice, including consent withdrawal, please see our Privacy Policy .

Reviewer #1: No

Reviewer #2: No

Reviewer #3: **Yes: ** Mohammed Elrabie Ahmed

---

## [Author Response · Author response to Decision Letter 2]

18 Jul 2025

Reviewer #1:

Authors have addressed all my previous comments and also provided clarity when the need arises. The manuscript has improved significantly.

Response 1:

Thanks again for reviewing our paper and sharing your valuable suggestions with us.

Reviewer #2:

General Comment:

Thank you for the opportunity to review the revised version of this manuscript. The authors have addressed many of the concerns raised in the initial review, and the paper has improved in clarity and rigor. Below I outline a few remaining issues and suggestions for further strengthening the manuscript.

Response:

Thank you for reviewing our paper again and sharing your insightful comments for improving the quality and readability of this protocol paper.

Comment 1: Introduction

The authors have effectively revised the introduction to include more background details and epidemiology on tinnitus. However, various sentences in the discussion of the “vicious cycle” remain disjointed and unclear. For example, “Untreated sleep disturbances can cause glymphatic dysfunction and brain atrophy as a result of accelerated cognitive decline in chronic tinnitus sufferers”. The structure of this sentence makes the cause effect relationship unclear implying that cognitive decline causes glymphatic dysfunction and brain atrophy rather than sleep disturbance—> glymphatic dysfunction-> brain atrophy-> cognitive decline. Please rework this section to clarify. The addition of more details about the DMN is effective and helps to clarify the study aims.

Response 1:

Thank you very much for pointing out this issue. After carefully checking, this part has been revised as “Of note, glymphatic dysfunction can lead to the accumulation of neurotoxic waste products in the brain, potentially contributing to neurological conditions, such as tinnitus and cognitive decline. Poor sleep quality may exacerbate this dysfunction, creating a vicious cycle in which chronic tinnitus disrupt sleep, and sleep disturbances further compromise glymphatic clearance. Thus, targeting this cycle through interventions that address both sleep disturbances and chronic tinnitus could hold significant therapeutic value, particularly in ageing populations.” On Page 4.Comment 2:

“Comorbidities: current clinical trials are lacking for the evidence of using TMS to manage sleep disturbances and chronic tinnitus simultaneously.” While the authors have added the related studies to the references, this sentence remains unclear. Please reword to clarify that current trials focus exclusively on one symptom, either tinnitus or sleep disturbance.

Response 2:

Thank you for this advice. This sentence has been re-written as “Comorbidities: current clinical trials only focus on one symptom (i.e., either tinnitus or sleep disturbances) and provide insufficient evidence to support the use of TMS for managing both sleep disturbances and chronic tinnitus simultaneously.” on Page 5.

Comment 3: Methods

Please include your justification and parameters for excluding severe hearing loss in the text. Please include details about data handling and security in the ethics section of the text. The authors have included a reference to justify their intensity parameters; however, a brief contextual explanation of this reference is lacking in the text. Please elaborate on how this reference supports your point. The addition of a subheading for safety and adverse events and the subsequent discussion of the adverse events checklist in the methods section is appreciated.

Response 3:

Thank you so much for pointing out the issues. In this study, we will check the medical records of the participants. Thus, if the participants are deaf or have severe hearing loss, they will be excluded from this study after screening. We have explained this issue on Page 8.

The data handling and security has been updated on Page 9 as “All participant data, including audiometric results and health surveys, were anonymized at collection and stored on password-protected computers. Access was limited to study investigators, and data retention follows IRB guidelines (5 years post-study).”

Yes. As you mentioned last time, we used the theta-burst TUS protocol in this trial. On Page 8, this part has been rewritten as “Theta-burst TUS (tbTUS) protocol will be used in this study for modulating neuronal activities and enhancing the neuroplasticity in hippocampal circuits [32,33]. The stimulation parameters of low-intensity TUS include Frequency: 5 Hz, pulse duration = 20 milliseconds, pulse repetition interval = 200 milliseconds and the total duration = 80 seconds, giving a total of 400 pulses.”. The reference list has been updated as well.

The subheading has been updated on Page 11.

Comment 4: Discussion

Enhanced discussion of targeting deeper brain structures with TUS is effective. Thank you for including a comprehensive limitations section. The authors now acknowledge limitations more clearly, which is appreciated.

Response 4:

Thank you for your feedback and comment. We have xxxx.

Comment 5:

The following are some language points that can be improved.

Till: use until (abstract)

“Toward effectively managing the co-occurring sleep disturbances and chronic tinnitus, it is fundamental for understanding the neural underpinnings of the vicious cycle” (intro)Suggest rephrasing for clarity (i.e. Understanding the neural underpinnings of the vicious cycle is fundamental to effectively managing co-occuring sleep disturbances and chronic tinnitus.)

“There are a bunch of studies” (intro) This phrase is too informal for scientific writing; consider replacing with “numerous studies have shown…”

Until to date: Revise to “To date,” or “until the present,” or “until now” (methods)

Response 5:

Thank you for carefully reviewing our paper. The points mentioned in the comment have been revised on Page 2, Page 4, Page 5 and Page 6.

Reviewer #3:

Comments have been addressed and answered completely in prefect way. The manuscript is well-structured and methodology well done No further comments.

Response:

Thanks again for reviewing our paper and sharing your valuable suggestions with us.

---

## [Editor Report · Decision Letter 2]

24 Jul 2025

Focused low-intensity hippocampal transcranial ultrasound stimulation (TUS) for sleep disturbances in patients with chronic tinnitus: A study protocol for a pilot randomized controlled trial

PONE-D-25-13152R2

Dear Dr. Lu,

We’re pleased to inform you that your manuscript has been judged scientifically suitable for publication and will be formally accepted for publication once it meets all outstanding technical requirements.

Kind regards,

Hantong Hu

Academic Editor

PLOS ONE
---

## [Editor Report · Acceptance letter]

PONE-D-25-13152R2

PLOS ONE

Dear Dr. Lu,

I'm pleased to inform you that your manuscript has been deemed suitable for publication in PLOS ONE. Congratulations! Your manuscript is now being handed over to our production team.

Kind regards,

on behalf of

Dr. Hantong Hu

Academic Editor

PLOS ONE